# Measuring Spatial Accessibility to Pick-Up Service Considering Differentiated Supply and Demand: A Case in Hangzhou, China

**Liyun Lin [1], Haoying Han [2],\*, Wanglin Yan [3] , Shun Nakayama [4] and Xianfan Shu [1]**

[1] Department of Land Management, Zhejiang University, Hangzhou 310058, China; linliyunZJU@zju.edu.cn (L.L.); shuxianfan@zju.edu.cn (X.S.)

[2] Institute of Urban and Rural Planning Theories and Technologies, Zhejiang University, Hangzhou 310058, China

[3] Faculty of Environment and Information Studies, Keio University, Fujisawa-City 2520882, Japan; yan@sfc.keio.ac.jp

[4] Graduate School of Media and Governance, Keio University, Fujisawa-City 2520882, Japan; shun-nakayama@keio.jp

\* Correspondence: hanhaoying@zju.edu.cn

**Abstract:** In recent years, customer pick-up at collection and delivery points has become a popular alternative to traditional home delivery, which is under great pressure. However, current service of pick-up facilities has seldom been geographically evaluated despite its general uneven distribution and diverse needs. In this paper, in order to interpret the differentiation in customers' service demands toward reception alternatives and in facilities' service excludability in different built environments, a two-step floating catchment area (2SFCA) method is improved to measure customers' spatial accessibility to pick-up facilities, providing a methodology to evaluate the match relation between the differentiated supply and demand of pick-up service. A case study of widespread automated parcel stations (APSs) is conducted in Hangzhou, China and correlative factors to residents' accessibility are discussed. From the results, residents' accessibility to pick-up service shows significant spatial unevenness and social inequity in the study area, which is found to correlate most to residences' maintenance management. As well-managed, gated communities generally hold effective access to exclusive services, most open communities and self-built, single houses are in need of improvement due to inadequate service stemming from a high aging rate, lack of property management, and low service availability of nonexclusive facilities in open areas.

**Keywords:** spatial accessibility; two-step floating catchment area method; last-mile delivery; demand differentiation; service excludability; gated communities

## 1. Introduction

There has been considerable expansion of e-commerce all over the world in recent years. With the rise of purchasing power, online security, and access channels, online B2C and C2C business has attained phenomenal growth in transaction volume and geographic coverage [1,2]. The consequent surging increase in parcel volumes, together with diverse delivery demands and the scattered distribution of online shoppers, have posed challenges for traditional home delivery (HD) during the terminal step of parcels to customers' doorsteps [3,4], making last-mile delivery the most expensive, most pollutive, and least efficient in all segments of the logistics system, which can account for 13%–75% of the total cost [5]. Meanwhile, although last-mile delivery plays an indispensably vital role in customers' online shopping experience, increasing problems stemming from inefficient service, such as failed

first deliveries, damaged parcels, inconvenient returns, and unqualified couriers, have become the bottleneck of e-commerce [6].

To solve last-mile problems, researchers, institutions, e-commerce companies (ECs), logistics service providers (LSPs), community service providers (CSPs), and many other stakeholders have eagerly worked together for decades and come up with diverse innovations relating to delivery routing, vehicles, and reception alternatives [7,8]. Customer pick-up at collection and delivery point (CDP), as a form of unattended reception without face-to-face contact with couriers [9], has been a hotspot in logistics research and practice as a sustainable, propagable, and widespread solution, with generally accepted advantages in economic efficiency [10,11], environmental friendliness [12,13], social values [14,15], and service quality [16,17]. Thus far, typical types of CDP include automated parcel stations (APSs), specialized manned ones (SM-CDP) such as post offices, and unspecialized manned ones (USM-CDP) such as convenient stores, which are becoming key features of e-commerce's and logistics players' strategies [18] and an important component in the delivery networks of urban areas in many countries, such as France, Germany, the UK, and Japan [19,20]. In China especially, the fast-growing CDPs have been included in planning public service facilities and have become a required service in residential and working units [21].

As researchers and providers generally focus on evaluating the benefits of customer pick-up in cutting operational and environmental costs, accessibility to CDPs, which is generally noted as having importance by customers [22] and found to be a positive influence on people's inclination regarding pick-up service and e-shopping frequency [23–25], is seldom geographically studied. Only a few researchers have studied the spatial distribution of pick-up services [18,26,27], but mostly under the assumption of homogeneous service and demand for all last-mile alternatives. This is despite great differences among customers' service preferences [28,29] and existing service exclusiveness at certain areas, which can lead to problems such as demand overestimation, low utilization, improper design of service network [30,31], and customer dissatisfaction [32,33].

As the concept of "pick-up" embeds customers into the delivery process, their diverse requirements toward various reception alternatives become an important issue in studying CDPs and their future layout. To fill the research gap, this paper aims to propose a method to more accurately measure the matching relation between differentiated demand and supply of pick-up service through accessibility evaluation. The following Section 2 will give a review on customers' differentiation in service demand among reception alternatives as well as facilities' differentiation in service excludability among multiform built environments and will introduce the improved two-step floating catchment area (2SFCA) method for accessibility evaluation, which can interpret the differentiation and realize a two-way match between demanders and providers. Section 3 presents the data processing and hypothesis proposing a case study of APSs in Shangcheng District, Hangzhou, China, which includes a user survey and geographical analysis. In Section 4, the results of the evaluation on residents' spatial accessibility to pick-up service and hypothesis testing in the study area are analyzed from the perspectives of spatial unevenness and social inequity in service provision, based on which conclusions and suggestions are made in Section 5, hoping to provide a reference for method application and local improvement.

## 2. Accessibility to Pick-Up Service: Considering Differentiated Demand and Supply

### 2.1. Preference Diversity towards Parcel Reception Alternatives

Alternatives of parcel reception mainly involve changes in receiving time, location, and behavior [34], compared to traditional HD. According to existing practices [35,36], SM-CDP and USM-CDP usually can provide manual services, including parcel inspecting, signing, returning, and cash-on-delivery at reception, but only at flexible times during limited working hours, while APSs can provide 24-hour electronic self-service but with a restricted number and size of container boxes. Aside from these functional values, CDP can also provide customers with financial and

emotional values [14,15] that largely depend on the local context [37–39]. Other alternatives include home reception boxes and home access systems, which provide similar functions to APSs but are uneconomical and rarely used [40].

For customers, their choice of all these alternatives comes from personalized demand on delivery service based on their consideration of various situational factors, such as location, security, reliability, time flexibility, speed, price, and function [41–45], which may differ with individual characteristics, such as shopping frequency, living conditions, working status, optimism, and innovation, according to previous research [46–49]. Some studies found that customers with regular work tend to receive parcels at the workplace or use CDPs near home, while the unemployed, freelancers, and retired individuals have more free time for pick-up or home delivery [50]. An investigation in the Netherlands showed that frequent purchasing online was connected to a higher probability of using collection points, which was more prevalent among females and larger households [51]. In addition, in terms of people's consideration for services, private car owners were found to care less about a CDP's location and more about its parking availability than walkers [42]. As the security issue was noted as an incentive for pick-up's popularity among females in China [52], different views were held by Swedish customers concerning risk of robbery [53]. Customers' choice between manned and unmanned CDPs may stem from their adaption to innovations and their need for social interaction and personal service [48,53].

## 2.2. Service Excludability Generating From the Built Environment

With regard to the design of a CDP network, the built environment can have a vital impact on the deployment and performance of CDPs by affording space, access, and activities. Some studies have proven that, in addition to internal factors such as population density and Internet access, CDP deployment is also affected by external factors such as the availability of deployment space, space-owners' willingness to join a network, transport infrastructure, and spatial accessibility to end-consumers [18], varying with the different designs of service points. As manned CDPs, similarly to retailers, often require easy access on open streets, APSs usually have a small occupation on an open subsidiary space to other constructs, such as at transportation nodes in Japan [20] and partner shops in France [18]. In China, however, the common location of APSs at the stilt floor or common area of residential, office, and commercial buildings [7] can lead to discrepancies in service excludability stemming from the wide distribution of gated communities in the country.

A gated community is usually considered a residential area that is enclosed by physical barriers such as walls, fences, or landscaping and is supervised by security personnel or electronic security systems [54,55], with restricted access not only to personal residences but also to the area's streets and neighborhood amenities [56]. Gated communities have been constantly criticized for aggravating social inequity and spatial fragmentation; researchers have argued that the efficient provision of urban services inside gated communities will lead to a diminished concern for the quality of services outside and the privatization of public space in gated communities could reduce the density and connectivity of urban road networks [57–59]. Previous studies on accessibility measurement have effectively studied differentiation among facilities' attractiveness and competitiveness but have seldom considered the impact of gated areas such as gated communities which, even when a facility is designed with open access, the physical and artificial barriers of gated areas can lead to the service exclusiveness of internal facilities and lengthened routes of relevant trips (as shown in Figure 1).

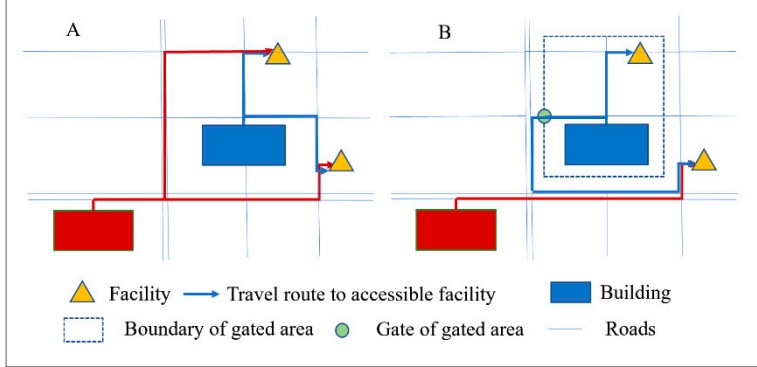

**Figure 1.** Impact of gated areas on facilities' service excludability and travel routes to facilities: (**a**) travel routes to accessible facilities without existence of gated area; (**b**) travel routes to accessible facilities under existence of gated area.

### 2.3. Research Model: Improved 2SFCA Method

The 2SFCA method is widely used in studies on spatial accessibility to public facilities. As time and financial budgets play no role in access to facilities with the same price and working time, 2SFCA is a suitable method for measuring accessibility to a certain type of CDP, which comprehensively considers the origin propulsion, destination attractiveness and proximity relationships among locations. As a special case of a gravity model of spatial interaction, the original 2SFCA method includes two steps [60]: first, assess the service availability of each provider as the ratio of service capacity to their surrounding demand within a threshold travel distance, which for CDPs indicates the average number of storage units available per time of demand within each's service area; second, calculate the service accessibility of each demander by summing up the availability ratios of providers within the same threshold travel distance, which stands for their accessible number of storage units available per time of demand.

To better interpret the differentiation among customers and facilities in this study, the 2SFCA method is modified as follows (as shown in Figure 2):

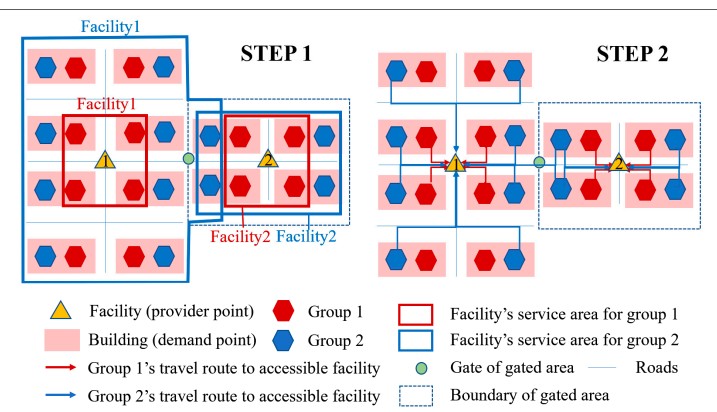

**Figure 2.** Two steps of catchment in the improved two-step floating catchment area (2SFCA) method.

For demand point i, where different groups of people {m} live, its overall accessibility to a certain type of CDP C is

$$A_i = \sum A_i(m) \times P_i(m) = \sum A_i(m) \times \frac{D_i(m)}{D_i}, \tag{1}$$

$$D_i(m) = N_i(m) \times (1 - P_0(m) \times P_{ad}(m) \times P_C(m) \times w(m)), \quad D_i = \sum D_i(m) \tag{2}$$

$$A_i(m) = \begin{cases} \sum_{j \in \{t_{ij} \leq t_0\}, j \notin G - G_i} R_j, & i \in G \\ \sum_{j \in \{t_{ij} \leq t_0\}, j \notin G} R_j, & i \notin G \end{cases} \quad , t_{ij} = \frac{d_{ij}}{v_m} \tag{3}$$

$$R_j = \begin{cases} S_j / \sum_{k \in \{t_{kj} \leq t_0\} \cap G_i} D_k(n), & j \in G \\ S_j / \sum_{k \in \{t_{kj} \leq t_0\}} D_K(n), & j \notin G \end{cases} \quad , t_{kj} = \frac{d_{kj}}{v_n} \tag{4}$$

where $A_i(m)$ is the accessibility of group m to C at i; $P_i(m)$ is the proportion of group m's demand in total demand for C at i; j is the provider point where a C is located; G and $G_i$ refer to all gated areas and the gated area that contains i respectively; $R_j$ is the provider-to-demanders ratio of j; $t_{ij}$ is the trip time of customers at locations i to j; $t_0$ is the threshold of trip time to provider point; $S_j$ is the capacity of parcel storage of C at j; k is the demand point with groups of people {n}; $d_{ij}$ is the distance between i and j; $v_m$ is the trip speed of customer m; $D_i(m)$ and $N_i(m)$ are the expected demand and the total populations of group m at i respectively; $P_0$, $P_{ad}$, and $P_C$ respectively refer to the percentage of people with zero reception, with preference of reception address at location i and in favor of C which vary among m; and w(m) is the average weekly reception of group m.

The improved method mainly involves changes in four aspects:

First, demand of expected customers for a certain service is differentiated among socioeconomic groups and integrated in the overall evaluation of accessibility (Formulas (1) and (2)). Based on previous research on customers' diverse preferences for final delivery (as mentioned in Section 2.1), we hypothesize that customers' parcel reception behavior is highly connected with their socioeconomic status (SES), which can be measured by three explanatory variables—reception frequency, address, and preferred service—which help define the demand structure among all reception alternatives.

Second, threshold of travel time and differentiated travel speeds are used to define the catchment area instead of threshold of travel distance (Formulas (3) and (4)). As the original 2SFCA only uses one catchment size for all populations, the threshold of travel time can better interpret the differences among people's mobility.

Third, exclusiveness of CDP service within gated areas is included (Formulas (3) and (4)). CDPs within gated areas are set as only available to people living or working inside and within a threshold of travel time for a more realistic simulation.

Finally, travel routes with detouring into or out of gated areas are imitated in the two-step catchment (Figure 2). For a more realistic route analysis, it is set that travel activities between buildings within gated areas and CDPs must go through the gate corresponding to the shortest total distance. In addition, difference in size of storage units is not considered in CDPs' capacity, since their design is hypothesized to conform to the frequency of various parcel sizes.

## 3. Case Study: Hypothesis, Methods, and Data

The Shangcheng District of Hangzhou, China, is selected as the study area in this research. As a center of social and economic activities in the south-central area of the city, Shangcheng District has been Hangzhou's core area throughout history as well as one of the earliest urban districts, bordering West Lake in the east and Qiantang River in the southeast. With a land area of 18 km², the district has a long-term residential population of 3.48 million and a working population of 2.42 million. It is also the most aging area, with 30.6% of population above 60, especially in the old town in the eastern part (Figure 3a). After years of evolution, Shangcheng District now possesses diversiform residential areas (Figure 3b) that vary in time of construction, spatial pattern, population density, and age composition, offering a typical case for studying diverse customer groups and built environments.

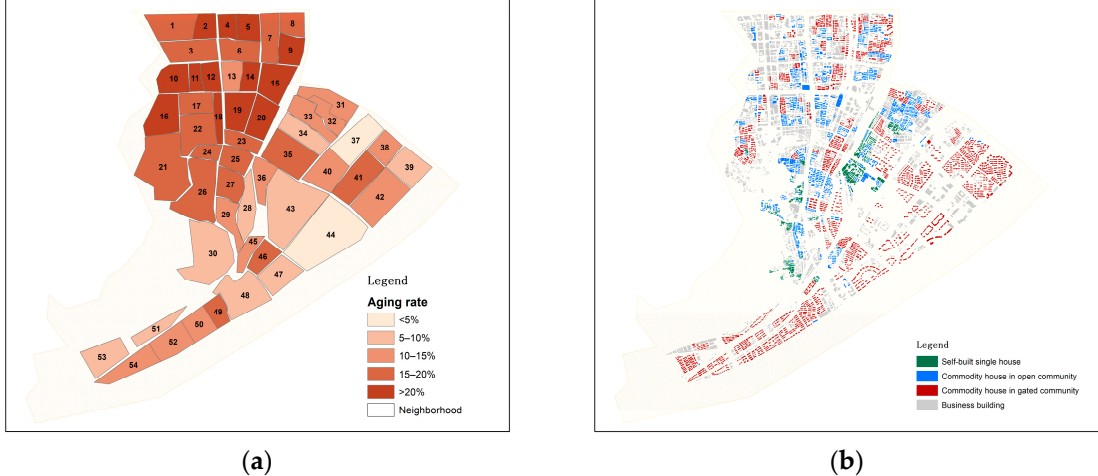

**Figure 3.** Distribution of aging population and housing types in the study area: (**a**) Aging rates (≥68) of neighborhoods; (**b**) residences and business buildings.

In addition, famous as the birthplace of the Alibaba Group and as "the city of e-commerce" in China, Hangzhou's significant growth in e-retailing and the parcel express industry in the past 20 years has promoted the emergence of various competing parcel reception alternatives, including HD, APS, SM-CDP, USM-CDP, and informal pick-up locations, which provide sufficient examples for this research. Considering the scant number of other types of CDPs, APSs are taken as the object of accessibility evaluation, which have an extensive distribution and a technically open design of utilization system in our study area.

The following sections will introduce the methods of data collection and processing during the application of the improved 2SFCA to our study area. All data except questionnaires were collected in July 2018.

### 3.1. Online and Ffield Survey: Investigating the Demand Differentiation of Customers

The following sections introduce the methods of data collection and processing during the application of the improved 2SFCA to our study area. All data except questionnaires were collected in July 2018. First, due to difficulties in obtaining adequate data from field investigations, an additional online survey, "Usage behavior and Service Preference of Parcel Pick-up Services" (Appendix A), was conducted from July to August 2018. The aim of this survey was to investigate people's general characteristics of parcel reception behavior and to provide a reference for estimating the demand of the whole population and defining catchment sizes in study area.

Our questionnaire consisted of 3 sections: Individual SES information, parcel reception behavior, and acceptance of pick-up trip. Parcel reception behavior included the 3 aforementioned aspects and acceptance of pick-up trip was expressed by accepted travel time and mode. SES variables included those with studied connections to reception behavior to explore people's demand differentiation, such as income, education, and housing type. A stratified random sampling approach was employed among our target groups (more than 12 years old according to a presurvey) so that links to the questionnaire were randomly spread on WeChat (the most popular social software in China), obtaining 657 responses, and 21 other responses were collected from a field survey on elders above 58 with a lower usage of WeChat. In total, 678 responses (Table A1) were answered in over 1 min for 31 questions and showed no significant difference between 20% from Hangzhou and 80% from other Chinese cities; thus, they are all valid for analysis.

In the questionnaire, the two main options for reception address were at/nearby home or the workplace. In addition, in order to reveal the potential demand concealed by current deficiencies in and compulsive or default use of CDPs, respondents' preferred service at the chosen address was selected

among HD, APS, SM-CDP, USM-CDP, and informal pick-up locations, based on a given introduction and self-perception rather than actual usage. As a reception alternative was always required due to commonly failed first home delivery and service restrictions, the first and second preferences were both considered in aggregated preference with a relative weight of 2:1. From the results (full details are given in Figure A1), 3 parameters showed no obvious relationship with each other but an obvious connection with gender and age:

1.  No significant difference in reception frequency was found between genders; average weekly reception generally decreased with an increase in age.
2.  As the choice of the majority (71.0%), home reception was preferred more by females and respondents of studying and retiring age than males and those of working age.
3.  For respondents preferring home reception, younger and female respondents were found to prefer APSs in their aggregated preference.
4.  For most respondents, the accepted travel time and travel mode to an APS was "within 2–5 min" and "on foot"; no age and gender differences were found.

### 3.2. Scenario Building: Estimating the Distribution of Expected Demand

First, the distribution of residents and workers in the study area was estimated. For residents, gender and age composition of the 2010 long-term population in 54 neighborhoods (data from the Sixth Nationwide Population Census) was adjusted with 2010–2018 birth/death rates for decomposing the 2017 long-term population (data from the 2011–2018 Hangzhou Yearbook (http://www.hangzhou.gov.cn/col/col805867/)). With household numbers of every community (data from house-renting website (hz.ke.com); some missing single houses were set with one household each), the average household size of each neighborhood was calculated based on which population was distributed, as households were allocated to each of the 3444 residential buildings in proportion to their total floor area (using building location/area/floor data from Baidu Map and E-dushi Map (hz.edushi.com)). For workers, the total population in 2017 was allocated to 3223 business buildings in proportion to their floor area, including all workplaces other than residences, such as office buildings, commercial centers, and government houses.

Second, based on age/gender discrepancies from the questionnaire analysis, expected APS customers for home and workplace reception are defined in Table 1 and spatially allocated according to the estimated distribution of residents and workers in Shangcheng District (Figure 4a).

**Table 1.** Definition of expected customers.

| Address | j (age) | j (gender) | $P_0$ | w | $P_{ad}$ | $P_C$ |
|---|---|---|---|---|---|---|
| | 13–17 | Male | 16% | 2.10 | 82% | 39% |
| | | Female | 8% | 1.75 | 93% | 43% |
| | 18–27 | Male | 4% | 1.75 | 60% | 34% |
| | | Female | 4% | 1.75 | 67% | 38% |
| | 28–37 | Male | 8% | 1.40 | 39% | 34% |
| | | Female | 4% | 2.10 | 44% | 38% |
| Home | 38–47 | Male | 10% | 1.75 | 48% | 34% |
| (differentiated) | | Female | 4% | 1.75 | 52% | 38% |
| | 48–57 | Male | 14% | 1.40 | 61% | 34% |
| | | Female | 4% | 1.40 | 61% | 38% |
| | 58–67 | Male | 38% | 1.05 | 100% | 34% |
| | | Female | 8% | 1.05 | 100% | 38% |
| | ≥68 | Male | 44% | 0.70 | 100% | 24% |
| | | Female | 40% | 0.70 | 100% | 28% |
| Home (undifferentiated) | ≥13 | All | 8% | 1.52 | 70% | 38% |
| Workplace | 18–57 | All | 7% | 1.66 | 54% | 34% |

### 3.3. Network Analysis: Measuring Residents' Spatial Accessibility to APSs

Network analysis of ArcGIS 10.2 was used for the two-step catchment in the improved 2SFCA and was conducted on existing road networks with a supplement of restrictions for detoured routes in/out of gated areas (data on roads, boundaries, and gates of gated areas from Baidu Map as shown in Figure 4b)) and 364 existing APSs (including those within 500 m of the study area) of 5 brands which were all technically open-access (location data from the mobile application of each brand in Table A2). As Table 2 presents, except for exclusive APSs in gated areas, catchment size was set as the most accepted 5-minute walk from the survey. Moreover, since APS capacity was similar among different brands, capacity for weekly reception was set as 588 boxes for all APSs in reference to a standard APS of 84 boxes from Fengchao, the most widespread brand in the study area. The original 2SFCA was also applied for comparison.

**Table 2.** Settings of analysis with improved and original 2SFCA.

| Parameters | Improved 2SFCA | Original 2SFCA |
|---|---|---|
| Service area | Trip mode: Walking<br>$t_0$ (min): 5<br>v (m/min): 83 (age 13–47)<br>70 (age 48–67)<br>54 (age ≥68) | $d_0$ (m): 415 |
| Travel route | With detour into/out of gated areas | No detour |
| Expected customers | With differentiated demand | With undifferentiated demand |
| Service availability | Inclusive in open areas and exclusive in gated areas | Inclusive in all areas |
| Facility capacity | $S_j$ (box/day): 588 | $S_j$ (box/day): 588 |

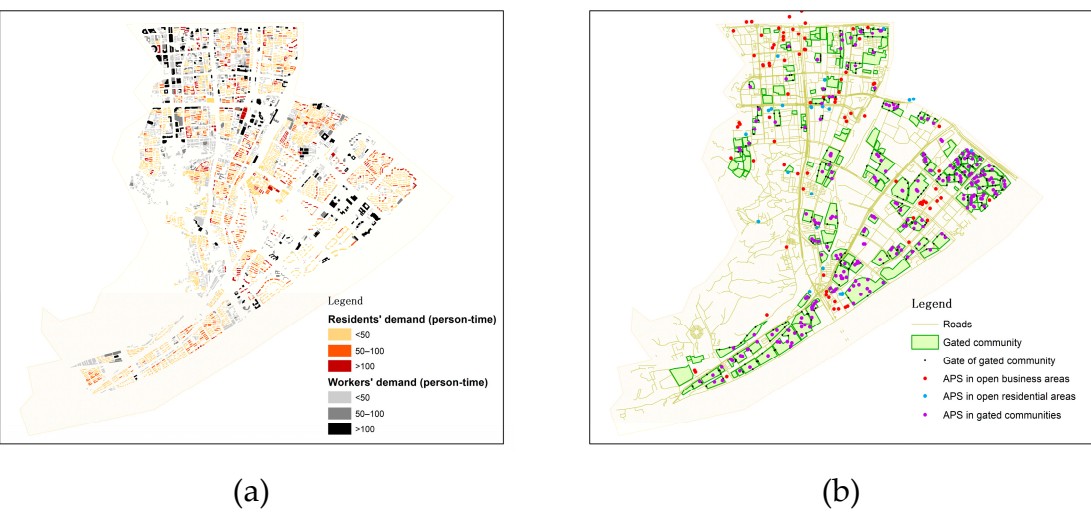

(a)          (b)

**Figure 4.** Geographical datasets of the study area for improved 2SFCA: (**a**) Distribution of demand of residents and workers for automated parcel station (APS) service; (**b**) distribution of roads, gated communities, and APSs.

### 3.4. Correlation Analysis: Explaining Spatial Unevenness

To explain the geographical discrepancy of accessibility to APS, 3 factors were proposed with a possible connection to service deployment and accessibility at certain residences, based on previous research findings and field observations:

1.  Demand adequacy. In studies of pick-up facilities in China and France [18,61], concentration in urban core areas and sparseness in peripheral areas stem mainly from regional difference in population density and preferred service. A sufficient number of accessible users is required for a cost-efficient pick-up facility with certain installation and operation costs, which is a main concern for providers in setting up or maintaining an APS within a certain area [18]. However, if serving excessive high-density demand, a facility's low availability can also result in overall low accessibility of surrounding customers.

2.  Space availability. Although APSs have high flexibility in deployment location, spatial differences still exist in the availability of appropriate spaces for installation. In our study area, since APSs are mostly located on the ground floor of residential, commercial, office, and other buildings, residents in nearby business areas may have locational advantages in reaching APSs with open access, as residential areas all share similar space usability.

3.  Maintenance management. As APSs occupy space in residential and business areas, rents are usually charged by property management agencies that also undertake daily maintenance for machine security and cleaning, while technical maintenance is provided by suppliers. While most gated communities are equipped with standard property management, self-built houses and open communities are more prone to management vacancy in our study area, which may become an impediment for the provision of internal facilities.

Thus, the following hypotheses were proposed: (H1) Local demand density has a positive correlation with deployment of APSs within residential areas; (H2) local demand density has a negative correlation with accessibility to APSs in residential areas with internal APSs; (H3) availability of nearby business areas is positively correlated with deployment of external APSs and accessibility to APSs in residential areas; and (H4) existence of standard property management has a positive correlation with deployment of internal APSs and accessibility to APSs in residential areas. The proposed hypotheses were tested with Spearman correlation analysis in SPSS 20, using the indicators for factors in Table 3, which were collected for each factor based on previously introduced data.

**Table 3.** Indicators of factors.

| Factors | Indicators | Indicator Collection |
|---|---|---|
| Demand adequacy | LDD: Local demand density | Density of expected demand for APS within Euclidean distance of 300 m [1] |
| Space availability | LB: Local business area | Footprint area of all business buildings within Euclidean distance of 300 m |
| Maintenance management | HT: Housing type | 1: Self-built single house 2: Commodity house in open community 3: Commodity house in gated community |
| Facility provision | IN: Internal APS | Number of accessible APSs (with best mobility) located in residential areas |
|  | EN: External APS | Number of accessible APSs (with best mobility) located in business areas |

[1] Officially recommended service radius of residential service facility in China [62].

## 4. Result Analysis

### 4.1. Spatial Accessibility to APSs Evaluated by Improved 2SFCA

The spatial accessibility of every customer to APS service in 3444 residential buildings was evaluated using the improved 2SFCA, given the 364 APSs in the study area. Residences' overall accessibility, A($\geq$13), was calculated by summing up the accessibility of three age groups with different mobility (13–47, 48–67, and $\geq$68), weighted by their proportion in total demand within the residence. The value of accessibility was graded as "low" (<0.5), "medium" (0.5–1.0) and "high" (>1.0). The spatial distribution of overall accessibility for every person and the time of demand in all residential buildings

is shown in Figure 5a, which presents obvious unevenness in that accessibility is high in most of the eastern part (neighborhoods 37–54), while other parts have large areas of low accessibility, especially for the middle part (neighborhoods 31–36), with no service coverage. According to Formula 1–4, spatial discrepancy of accessibility can be generated from the following three aspects: Service availability (provider-to-demanders ratio) of accessible APSs, number of accessible APSs, and age composition of demand location.

The first one proves to be a key determinant of accessibility, as most buildings with medium or high accessibility are located near APSs, with an R value above 1.0, especially for those in gated areas (Figure 5b). APSs in gated communities are exclusive to residents within each community, which is generally smaller than other APSs' service areas, resulting in their smaller serving population and the highest average R of 2.9. APSs in open residential and business areas are available to every resident and worker within a five-minute walk, so their R is commonly below 1.0 because of the large quantity of accessible demand, which, for facilities in business areas, is relatively smaller to cover the demands of more densely distributed workers.

The number of accessible APSs presents a limited impact on residents' accessibility to APSs. In total, 57.1% of serviced customers and 80.8% of serviced areas (with best mobility) are accessible to more than one facility (Figure 5c). As a high percentage of existing APSs are located in gated communities (Table 4), 119 of 173 gated communities contain internal APSs, of which 92.4% are covered by an overlapping service, since 65 possess more than one APS and more than half can access APSs outside, conforming to mostly medium or high accessibility in gated communities. However, many other serviced buildings in neighborhoods 1–30 remain low in accessibility even with overlapping service, which stems from the low provider-to-demanders ratio of APSs in open areas.

**Table 4.** Service availability and coverage of APSs by location in the study area.

| Location of APSs | Number | Demand Coverage | Average R |
|---|---|---|---|
| APSs in gated communities | 254 | 45.6% | 2.9 |
| APSs in open residential areas | 24 | 21.6% | 0.7 |
| APSs in open business areas | 86 | 34.3% | 0.5 |
| Total | 364 | 73.8% | 2.1 |

Age composition of demand location can contribute to unevenness in residents' accessibility among age groups. Older residents generally have lower accessibility and less demand coverage. As Figure 5e shows, the elderly above 47 have 6.9%–13.3% less coverage on their demand due to their limited mobility. The location of most age-limited service areas (yellow and blue areas in Figure 5d) in an area with a high aging rate above 15% (neighborhoods 1–29) is another reason. Thus, a high proportion of the elderly are inaccessible to APS and this can result in a substantial decrease in a location's overall accessibility $A(\geq 13)$ from a higher one for youths, a prominent group in the western part of the study area (Figure 5f).

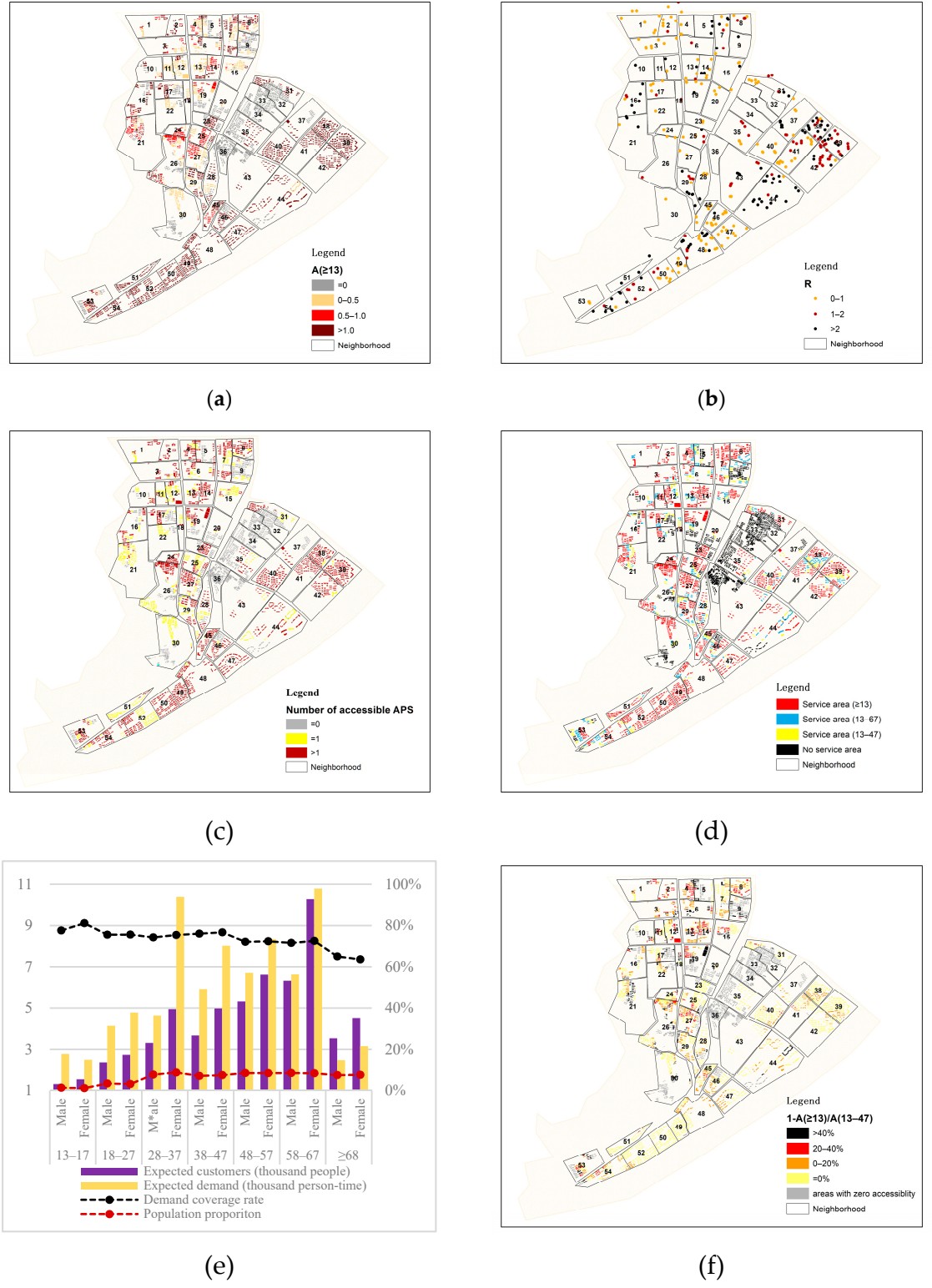

**Figure 5.** Results of accessibility evaluation in the study area (improved 2SFCA): (**a**) Overall accessibility of residences; (**b**) R of all APSs; (**c**) number of accessible APSs for residents (13–47); (**d**) service coverage of all APSs for residents; (**e**) demand coverage of all APSs for residents; (**f**) reduction from accessibility of residents aged between 13 and 47 to the overall accessibility of residences.

### 4.2. Comparison between Improved and Original 2SFCA

While measured by the original 2SFCA with homogeneous demanders and providers (Figure 6), the demand of most groups is overestimated and the limited mobility of the elderly is neglected, leading to an overall overvalued demand coverage and undervalued accessibility. Compared to the improved 2SFCA, for the eastern part with high concentration of APSs, most residences comprising mainly gated communities still offer good accessibility. However, the accessibility of gated communities in the western and middle areas decreases, along with sharing their internal APSs with surrounding residents. By comparison, the improved 2SFCA embodies validity in:

1.  Estimating the distribution of service demand which varies among customer groups;
2.  Defining facilities' service areas based on differences in facilities' excludability and customer mobility;
3.  Evaluating service availability and accessibility with an overall consideration for different demand groups.

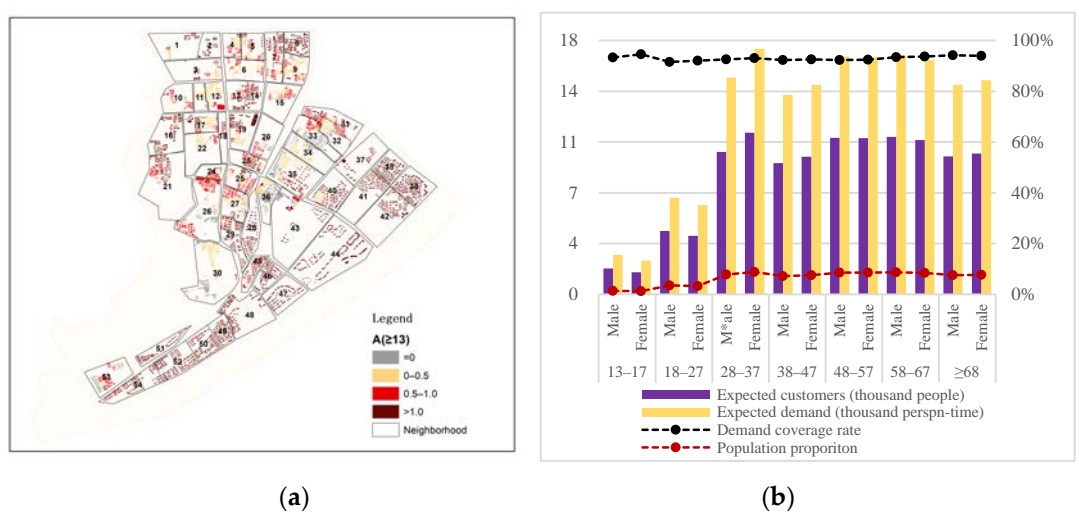

| (**a**) | (**b**) |
| --- | --- |

**Figure 6.** Results of accessibility evaluation in the study area (original 2SFCA): (**a**) Overall accessibility of residences; (**b**) demand coverage of all APSs for residents.

### 4.3. Correlation Analysis

From Table 5, results verify hypothesis H4, as HT is moderately and positively correlated to A(≥13) and IN, with coefficients of 0.496 and 0.585 respectively, while H1 is invalid, since LDD shows no correlation with IN. LDD shows a moderate negative correlation ($r_s$ = −0.281) with A(≥13) for a residential area with internal APS, providing support for H2. For H3, LB shows a positive correlation with EN in accord with the hypothesis. Correlation between LB and A(≥13) is negative for all residences in contrast to H3, but is positive yet weak for residences without an internal APS, which may stem from external APSs' weak correlation and contribution to a residence's overall accessibility compared with internal ones. Thus, maintenance management is validated with a positive correlation to internal deployment and accessibility to APSs for residential areas, while demand adequacy and space availability's correlation to APS accessibility is only partly validated.

**Table 5.** Results of correlation analysis.

| | | A (≥13) | | IN | | EN | |
|---|---|---|---|---|---|---|---|
| | | $r_s$ | Sig. | $r_s$ | Sig. | $r_s$ | Sig. |
| IN | | 0.502 | 0.000 | \ | \ | 0.100 | 0.000 |
| EN | | 0.306 | 0.000 | 0.100 | 0.000 | \ | \ |
| HT | | 0.496 | 0.000 | 0.585 | 0.000 | −0.043 | 0.012 |
| LB | Total | −0.329 | 0.000 | −0.221 | 0.000 | 0.205 | 0.000 |
| | IN = 0 | 0.137 | 0.000 | −0.062 | 0.011 | 0.219 | 0.000 |
| | IN≥ 1 | −0.290 | 0.000 | −0.041 | 0.082 | 0.391 | 0.000 |
| LDD | Total | −0.173 | 0.000 | 0.006 | 0.742 | 0.150 | 0.000 |
| | IN = 0 | −0.057 | 0.020 | 0.141 | 0.000 | 0.038 | 0.123 |
| | IN≥ 1 | −0.281 | 0.000 | −0.119 | 0.000 | 0.317 | 0.000 |

Notes: IN-internal APS; EN-external APS, LB-local business area; LDD-local demand density; HT-housing type; n = 3444 (total); n = 1659 (IN = 0); n = 1785 (IN ≥ 1).

*4.4. Mode Generalization*

Based on the correlation between factors, 10 modes can be summarized with different combinations of housing type, relative location to business center, possession of internal APS, and residence accessibility (Table 6).

**Table 6.** 10 modes of accessibility of residential buildings.

| Mode | Housing Type | Proximity to Business Center | Possession of Internal APS | Accessibility to APS |
|---|---|---|---|---|
| 1 | | √ | √ | High |
| 2 | Gated community | × | √ | Medium–high |
| 3 | | √ | × | Low |
| 4 | | × | × | Low (zero) |
| 5 | | √ | √ | Medium |
| 6 | Open community | × | √ | Medium |
| 7 | | √ | × | Low–high |
| 8 | | × | × | Low (zero) |
| 9 | Self-built single | √ | × | Low |
| 10 | house | × | × | Low (zero) |

Notes: "√" refers to being in proximity to business center/possessing internal APS; "×" refers to not being in proximity to business center/possessing no internal APS.

Considering service equity in all age groups, areas with overall low accessibility (A(≥13) <0.5) and a significant reduction (≥20%) from A(13–47) to A(≥13) are both defined as inadequately serviced areas that require improvement in service provision. As shown in Figure 7, these poorly serviced areas mostly lie in the west of the study area, with a high frequency of Modes 3, 4, and 7, and in the middle mostly occupied by Modes 8 and 10. The majority of open communities and self-built single houses retains low accessibility without internal APSs, especially in the middle, which is far from the business center and lacks an inclusive APS. In addition, almost all gated communities occupying the western area belong to Modes 1 and 2, with high accessibility, as opposed to eastern gated communities, which generally can only reach an APS at a nearby business area.

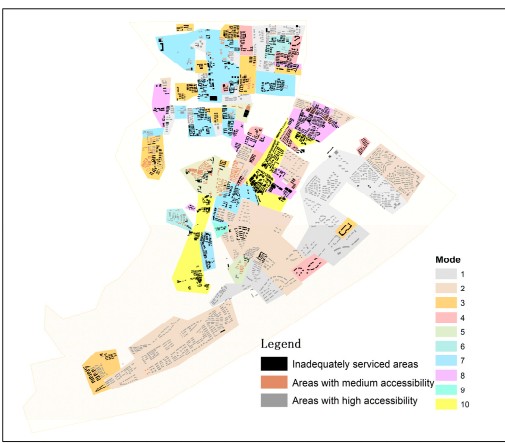

**Figure 7.** Inadequately serviced area and mode distribution in the study area.

## 5. Conclusions and Discussion

A growing and diversifying demand for parcel reception reflects changes in people's lifestyles, showing that online shopping is now an essential means of people's daily consumption. As pick-up services meet people's need for time and location flexibility, they have characteristics of a public service by consolidating goods of many and unspecified consumers [8]. Thus, it is necessary to evaluate pick-up services from the aspect of social need and with a concern for differentiation, which can help improve both the effectiveness and efficiency in the future arrangement of logistics services.

This study provides a methodology to evaluate the match relationship between differentiated demand and supply of pick-up services in a Chinese context, where differentiation mainly stems from the diversity in customer preference and the exclusivity of gated communities. An improved 2SFCA method is established to evaluate customer accessibility to pick-up facility by comprehensively analyzing both sides of supply and demand, which has been shown to be capable of managing differences in facilities' excludability and customer demand and mobility. Aside from our case study on APSs, this method is also replicable in other types of pick-up facilities or other regions and hopefully can provide a reference for further service evaluation to providers and planners.

In our case study on the residential areas of Shangcheng District, Hangzhou, the spatial unevenness and social inequity of residents' accessibility to APSs is identified, with the strongest discrepancy among housing types, which varies in the maintenance management of the facility. As gated communities are more favored in the deployment of APSs, they mostly retain good accessibility to pick-up services, excluding a small number of internal residents, despite lowered serviceability by high-demand density. Open communities and self-built single houses, on the contrary, have not only a scarce distribution of internal facilities, but also a high proportion of aging residents with low mobility. Facilities in open business areas also contribute to service coverage of these areas, as proximity to business areas shows a moderate positive correlation to service accessibility; however, this can hardly compensate for the vacancy of internal APSs due to the large number of serviced customers. Thus, open communities and self-built single houses concentrated in the middle and western areas generally have low accessibility to APS service.

Low accessibility in these open communities and self-built single houses may stem from their early construction and different land property, where vacancy of property management, unsecured environment, high aging rates and uncertainty of land renewal can all make providers wince. These areas with deficient service urgently require improvements in service supply, since it is not only an issue of equity for the elderly but also a critical problem for community sustainability that can result in less attraction for young residents and an exacerbation of aging problems. Since possession of an internal facility is vital to sufficiently service residential areas, the integration of pick-up services into current public facilities such as bus stations and community service centers, upgrading service in traditional logistics terminal facilities such as post offices, and promoting the importance of pick-up

facilities in urban planning can be helpful measures to overcome the obstacles of service provision in these areas.

Our online and field surveys also reveal a current problem with the compulsive use of CDPs, given the complaints from 44% of respondents, which also requires urgent improvement in the standard operation of pick-up services. This problem can also result in lower accessibility to APSs in their actual utilization, with the inclusion of customers preferring other services compared to the evaluation results in the paper.

It is also necessary to point out that, though the improved 2SFCA method in this paper is better at simulating the real-world demand for and operation of pick-up services compared to previous methods that do not consider relevant differentiation, the assessment results in this case study are still insufficiently accurate because of the lack of smaller scale data on residents' actual demand, facilities capacity, and population distribution. While the operation of pick-up services can be standardized without compulsion in the future, data on the practical utilization of pick-up facilities will be very helpful in obtaining a spatial portrait of the regional demand for and enhancement of evaluation accuracy. Distance decay of accessibility is also not included in this study, given the similar satisfaction within the small and walkable service radius of the terminal service facilities in the urban areas of Chinese cities, which may be unsuitable for studies in regions with a heavy reliance on car travel.

**Author Contributions:** Conceptualization, L.L., H.H., and S.N.; methodology, L.L.; software, L.L.; validation, L.L.; formal analysis, L.L.; investigation, L.L.; resources, L.L.; data curation, L.L. and X.S.; writing—original draft preparation, L.L.; writing—review and editing, H.H.; visualization, L.L.; supervision, W.Y.; project administration, L.L.

**Funding:** This research was funded by the National Natural Science Foundation of China, grant number 51778560.

**Acknowledgments:** I would like to express my gratitude to all my fellow students for their great help in collecting questionnaires in this study, including Yisong Peng, Xiaodong Zhang, Rui Wang, Bingqin Xu, Xinyu Kong, Junying Luo, Yu Chen and Beiqi Wang. I am also grateful to my parents who kept encouraging me to work on this study.

**Conflicts of Interest:** The authors declare no conflict of interest.

## Appendix A. Questionnaire on Usage Behavior and Service Preference of Parcel Pick-Up Services (Questions on Parcel Sending Are not Included)

Hello! We are a research group from Department of land resources management of Zhejiang University. We invite you to participate in this survey on usage intention of Parcel Pick-up Services.

Parcel Pick-up is a reception alternative emerged in recent years, which have advantages of time flexibility, privacy of personal information and security of reception. At present, pick-up facilities in China mainly include parcel lockers, specialized pick-up points, chain convenience stores and private stores. Specialized pick-up points such as Cainiao Station and Mama Station generally have manned pick-up and sending service, with operation hours between 8:00–18:00. Parcel lockers such as Sudiyi Locker and Fengchao Locker have 24 h self-service of pick-up and sending. Convenience stores have manned pick-up and sending services, running 24 h a day. And private stores have manned pick-up service, with varied business hours that is usually before 22:00.

The data obtained in this survey will be helpful to future construction of urban logistics facilities and to improvement of your life quality. We will keep your answers confidential in accordance with the Statistics Law. Thank you for your cooperation.

1. What is your gender?
A. Male                                             B. Female
2. What is your age?
A. 17 years old and below                          E. 48–57 years old
B. 18–27 years old                                 F. 58–67 years old
C. 28–37 years old                                 G. 68 years old and above
D. 38–47 years old

3. What is your educational background?
A. High Schools (secondary Schools) and below

B. Associate bachelor C. Bachelor

D. Master E. PhD

4. How about your monthly income (after tax)?

A. Less than 3000 RMB D. 9000–12,000 RMB

B. 3000–6000 RMB E. Over 12,000 RMB

C. 6000–9000 RMB

5. Your current working status?

A. Students C. Employed

B. Unemployed

6. Have you ever used pick-up service?

A. Yes B. No

7. How many times do you receive parcels per week on average?

A. Almost zero D. 3–4 times

B. Less than 1-time E. 5–6 times

C. 1–2 times F. More than 6 times

8. What type of residence are you living in?

A. Self-built single house

B. Commodity house in open community

C. Commodity house in gated community

D. Collective dormitory E. Other________

9. What are the existing alternatives of receiving parcels nearby your residence? [multiple choice questions]

A. Manual home delivery

B. Parcel lockers

C. Specialized pick-up points

D. Chain convenience stores

E. Private stores

F. Informal locations

(If students or unemployed, questions 11–13 need not be filled in)

10. What type of workplace do you work in?

A. Office building

B. Stores

C. No fixed place of work

D. Other________

11. What are existing alternatives of receiving parcels near your workplace? [multiple choice questions]

A. Manual delivery to workplace

B. Parcel lockers

C. Specialized pick-up points

D. Chain convenience stores

E. Private stores

F. Informal locations

12. How do you usually go to and from your place of work and residence?

A. On foot C. By bicycle or motorcycle

B. By bus or Subway D. Private car E. Other______

13. Where do you prefer to receive parcels?

A. Near/at home (please answer questions 14–19)

B. Near/at workplace (please answer questions 20–25)

C. Other______ (please answer questions 26–27)

14. What are the main reason you choose to collect parcel near/at home? [multiple choice questions]

A. Inconvenient collection in workplace D. More privacy

B. Near to bring it home E. Able to receive in nonwork days

C. Able to be collected by family F. Other________

15. What time is usually convenient for you to collect parcel near/at home? [multiple choice questions]

A. Work days 8:00–18:00

D. Weekends 8:00–18:00

B. Work days 18:00–22:00

E. Weekends 18:00–22:00

C. Work days after 22:00

F. Weekends after 22:00

16. Please choose at least two alternatives of reception that you would like to use at home and rank them according to your preferences.

A. Manual home delivery

B. Parcel lockers

C. Specialized pick-up points

D. Chain convenience stores

E. Private stores

F. Informal locations

17. What are your main considerations for choice above? [multiple choice questions]

A. Closeness to home

E. Professional manned assistance

B. Privacy and personal security

F. Accessorial services

C. Parcel security

G. Convenient transportation

D. Time flexibility

18. From your home, what is the longest time you can afford to go to pick-up facility?

A. Within 2 minutes

D. 8–11 minutes

B. 2–5 minutes

E. More than 11 minutes

C. 5–8 minutes,

19. From your home, which mode of transportation can you accept to go to pick-up facility? [multiple choice questions]

A. On foot

C. By bicycle or motorcycle

B. By bus or Subway          D. Private car

E. Other______

20. What are the main reason you choose to collect parcel near/at workplace? [multiple choice questions]

A. Inconvenient collection at home

D. More privacy

B. Near to bring it to workplace

E. Able to receive in working hours

C. Able to be collected by coworkers

F. Other________

21. What time is usually convenient for you to collect parcel near/at workplace? [multiple choice questions]

A. Work days 8:00–18:00

D. Weekends 8:00–18:00

B. Work days 18:00–22:00

E. Weekends 18:00–22:00

C. Work days after 22:00

F. Weekends after 22:00

22. Please choose at least two alternatives of reception that you would like to use at workplace and rank them according to your preferences.

A. Manual delivery to workplace

B. Parcel lockers

C. Specialized pick-up points

D. Chain convenience stores

E. Private stores

F. Informal locations

23. What are your main considerations for choice above? [multiple choice questions]

A. Closeness to home

E. Professional manned assistance

B. Privacy and personal security

F. Accessorial services

C. Parcel security

G. Convenient transportation

D. Time flexibility

24. From your workplace, what is the longest time you can afford to go to pick-up facility?

A. Within 2 min

D. 8–11 min

B. 2–5 min

E. More than 11 min

C. 5–8 min,

25. From your workplace, which mode of transportation can you accept to go to pick-up facility? [multiple choice questions]

A. On foot

C. By bicycle or motorcycle

B. By bus or Subway          D. Private car

E. Other______

26. Which alternatives do you use to receive parcels in the place you filled in? [multiple choice questions]
A. Manual delivery to door
B. Parcel lockersC. Specialized pick-up points
D. Chain convenience stores
E. Private stores
F. Informal locations
27. When do you usually receive parcels in the place you filled in? [multiple choice questions]
A. Work days 8:00–18:00 D. Weekends 8:00–18:00
B. Work days 18:00–22:00 E. Weekends 18:00–22:00
C. Work days after 22:00 F. Weekends after 22:00
28. Do you agree with the following problems in existing pick-up service?
A. The courier put parcels at pick-up facilities without permission.
B. Inadequate pick-up facilities
C. Inconvenient transportation to pick-up facilities
D. Manned service at pick-up facilities is not professional
E. Unreasonable charges for pick-up service
F. Other issues_________
G. No problem in pick-up service

## Appendix B

**Table A1.** Profile of respondents.

| Variable | n | Percentage | Variable | n | Percentage |
|---|---|---|---|---|---|
| Gender | | | Education level | | |
| Male | 306 | 45.1% | Associate bachelor and below | 227 | 33.5% |
| Female | 372 | 54.9% | Bachelor | 253 | 37.3% |
| Age | | | Master and above | 198 | 29.2% |
| 13–17 | 139 | 20.5% | Monthly income | | |
| 18–27 | 184 | 27.1% | <3000 | 135 | 19.9% |
| 28–37 | 149 | 22.0% | 3000–6000 | 148 | 21.8% |
| 38–47 | 66 | 9.7% | 6000–9000 | 102 | 15.0% |
| 48–57 | 94 | 13.9% | 9000–12,000 | 99 | 14.6% |
| 58–67 | 21 | 3.1% | >12,000 | 110 | 16.2% |
| ≥ 68 | 25 | 3.7% | Housing type | | |
| Working status | | | Self-built single house | 55 | 8.1% |
| Employed | 409 | 60.3% | Commodity house in open community | 83 | 12.2% |
| Unemployed | 269 | 39.7% | Commodity house in gated community | 369 | 54.4% |
| Student | 216 | 31.9% | Collective dormitory | 171 | 25.2% |
| Experience of pick-up reception | | | | | |
| Yes | 609 | 89.8% | | | |
| No | 69 | 10.2% | | | |

**Table A2.** Existing APSs in the study area.

| Brand | Provider type | Number |
|---|---|---|
| Fengchao | Logistics service provider | 280 |
| Sudiyi | Logistics service provider | 44 |
| Lejia | Community service provider | 19 |
| Gegexiaoqu | Community service provider | 14 |
| Jingdong | Logistics service provider | 7 |
| | Total | 364 |

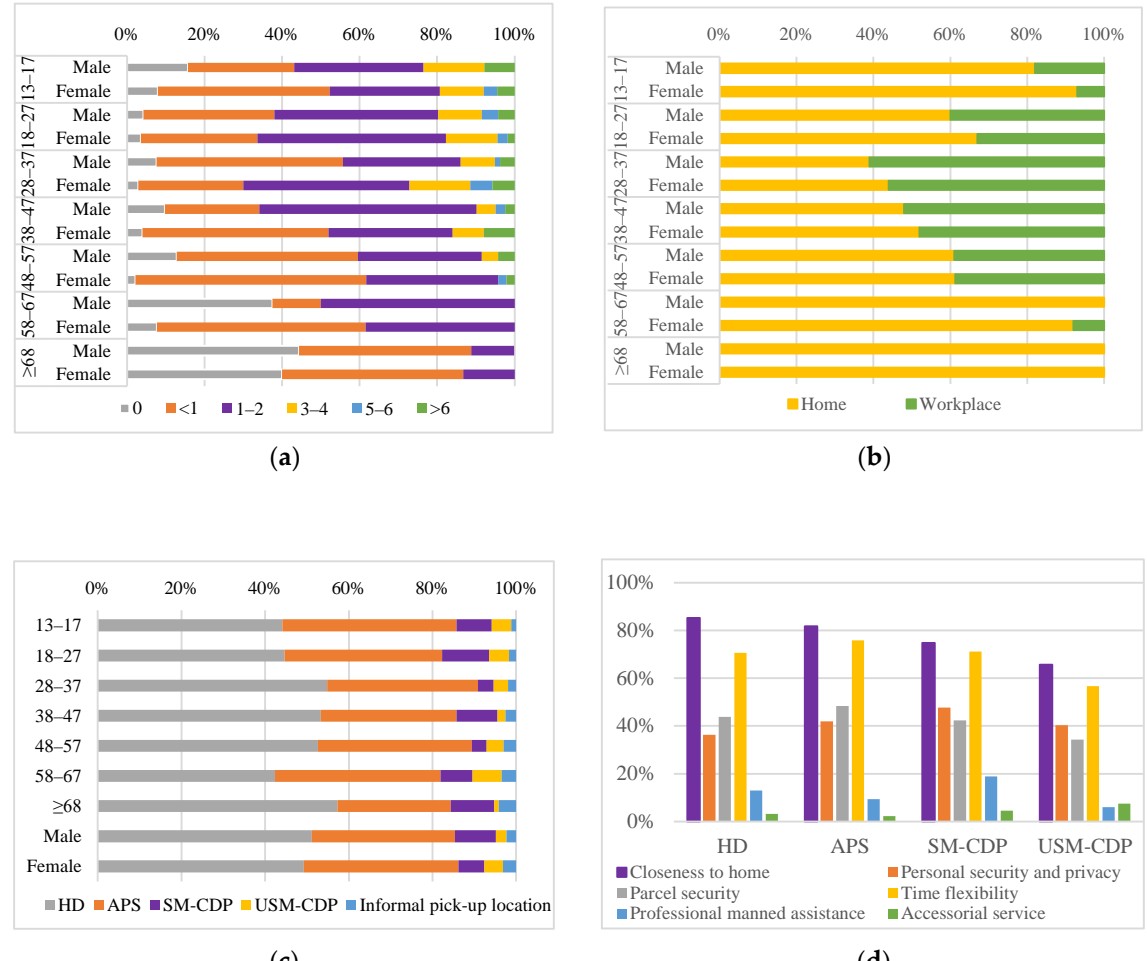

**Figure A1.** Results of questionnaires: (**a**) Weekly reception frequency; (**b**) preferred reception address; (**c**) aggregated service preference for home reception (weighted sum method is used in calculating the aggregated preference to eliminate the difference of gender composition and age composition among respondents); (**d**) considerations for home reception service.

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
