# Peer review of "Measuring Spatial Accessibility to Pick-Up Service Considering Differentiated Supply and Demand: A Case in Hangzhou, China"

_sustainability, doi:10.3390/su11123448_

Round 1

Reviewer 1 Report

In my opinion, the paper is very good and well explained (the introduction provide sufficient background and include all relevant references, the results clearly presented, and English is good).

I think that the main strength relies in the fact that all the process has started with a survey, in order to assess the opinion of the actual users of the services, thus providing a solution with a higher potential of market acceptance.

Furthermore, I think that the provided solution is easily scalable, at least in the Chinese environment.

Author Response

Thank you so much for such positive comment!  They have important guiding significance for my future research work.

Reviewer 2 Report

The article deals with a very current and important problem that occurs in many cities around the world. The problem of the last mile is growing year by year, especially since shopping is more and more often made online. As a result, there is an increasing personalization of deliveries and, as a consequence, more frequent and faster deliveries. This is a huge challenge for the logistics sector. One of the solutions that reduce congestion in the city are deliveries to collection and delivery points. It happens, however, that the location of points is not suited to the needs of customers. In this paper, the authors proposed an improved two-step floating catchment area (2SFCA) in order to measure customers' spatial accessibility to pick-up facilities, including supply and demand of pick-up services. In this respect, the authors fill the research gap.

The paper is written at a very good level. The structure of it is correct, and the presentation of results from the conducted research is transparent and clear.

Therefore, my comments only concern minor shortcomings at work:

In introduction it could be good to present general overview on the paper’s content (section 1, 2 etc.)

p. 4 there is no space between word "follows" and "(as shown in...)

p. 4. p. 4 “Based on previous research on customers’ behaviors and preferences of final delivery, we hypothesize that customers’…” -  what kind of research? There is no source confirming your statement (it should be given the source)

p. Section 5. 3. Case study: hypothesis, methods and data shouldn’t start with the figures.

Author Response

Thank you so much for such positive comment!  They have important guiding significance for my future research work.

I have made revisions to my work according to your comments. Except those in figure setting and spelling, general overview is added in intriduction which does help clear my paper content. And in p.4, I mentioned "based on previous researches" two times without details, which now have been clarified to refer to researches mentioned in section 2.1(literature review on customers' diverse preferences in parcel reception).

Thanks again for your advice!